# Mental Health States Experienced by Perinatal Healthcare Workers during COVID-19 Pandemic in Italy

**DOI:** 10.3390/ijerph18126542

**Published:** 2021-06-17

**Authors:** Loredana Cena, Matteo Rota, Stefano Calza, Barbara Massardi, Alice Trainini, Alberto Stefana

**Affiliations:** 1Observatory of Perinatal Clinical Psychology, Department of Clinical and Experimental Sciences, Section of Neuroscience, University of Brescia, 25123 Brescia, Italy; alice.trainini@unibs.it (A.T.); alberto.stefana@gmail.com (A.S.); 2Unit of Biostatistics and Biomathematics & Unit of Bioinformatics, Department of Molecular and Translational Medicine, University of Brescia, 25123 Brescia, Italy; matteo.rota@unibs.it (M.R.); stefano.calza@unibs.it (S.C.); 3Istituto Zooprofilattico Sperimentale della Lombardia e dell’Emilia Romagna (IZSLER), 25124 Brescia, Italy; barbara.massardi@unimi.it; 4Department of Clinical Sciences and Community Health, University of Milan, 20122 Milan, Italy

**Keywords:** COVID-19, pandemic, depression, anxiety, stress, burnout, healthcare workers, correlators

## Abstract

Background: The ongoing COVID-19 pandemic has had an impact on mental health status in a variety of populations. Methods: An online non-probability sample survey was used to assess psychological distress symptoms and burnout among perinatal healthcare professionals (PHPs) during the pandemic in Italy. The questionnaire included the Depression, Anxiety, and Stress Scale-21 (DASS-21) and the Maslach Burnout Inventory (MBI). Demographic and occupational factors associated with stress, anxiety, and depression symptoms were analyzed. Results: The sample size was 195. The estimated self-reported rates of moderate to severe anxiety symptoms, depression symptoms, and perceived stress levels were 18.7, 18.7, and 21.5%, respectively. Furthermore, 6.2% of respondents reported burnout. One factor associated with all three self-reported psychological distress issues was suffering from trauma unrelated to the pandemic (aOR: 7.34, 95% CI: 2.73–20.28 for depression; aOR: 6.13, 95% CI: 2.28–16.73 for anxiety; aOR: 3.20, 95% CI: 1.14–8.88 for stress). Compared to physicians, psychologists had lower odds of developing clinically significant depressive symptoms (aOR: 0.21, 95% CI: 0.04–0.94) and high stress levels (aOR: 0.19, 95% CI: 0.04–0.80). Conclusions: High rates of self-reported symptoms of depression and anxiety, as well as perceived stress, among PHPs were reported during the COVID-19 pandemic. Health authorities should implement and integrate timely and regular evidence-based assessment of psychological distress targeting PHPs in their work plans.

## 1. Introduction

The coronavirus disease (COVID-19) pandemic has thus far infected more than 100 million people and caused almost three million deaths globally (see https://covid19.who.int/ (accessed on 16 May 2021). In an effort to contain the spread of infection, governments across the world have been imposing mitigation strategies [1], which have caused side effects such as physical and emotional isolation, huge economic losses, and disrupted healthcare services. All of these have led to a global atmosphere of uncertainty and psychological distress [2,3,4].

During this global emergency, a growing body of studies have documented the impact of this situation on mental health status and related risk factors of vulnerable populations, such as people with mental disorders [5,6] and front-line healthcare workers [7,8]. Concerning the latter, the available literature has been consistent in reporting that healthcare professionals are currently under tremendous pressure, resulting especially from increased workload, increased risk of infection and uncertainty about the efficacy of available (off-label) treatments for COVID-19, and a lack of in-person social contacts with friends and relatives [9,10]. As a result, these professionals have developed mental health problems such as anxiety, depression, rage, denial, somatization, insomnia, and burnout [11,12], which could severely affect their work performance (including attention, understanding, and decision-making capability) and, more broadly, their overall well-being [13]. Of concern, a specific subsample of healthcare workers, perinatal healthcare professionals (PHPs), is receiving less attention than it deserves.

Working with pregnant women and babies can elicit intense emotional responses [14,15,16,17] that, if not managed properly, may have the potential of further adversely affecting the quality of PHPs’ healthcare work. Studies that analyzed the relationship between emotion and, for example, decision-making have indicated that the emotions experienced by healthcare workers can induce an emotional bias in decision-making which, in turn, can result in errors and adverse events [18,19]. Other studies have found that physicians’ emotional responses can negatively influence medical safety [20,21,22] as well as patient safety [22]. A recent study on the influence of pediatricians’ emotional factors on pediatric medical adverse events (based on reports from a Japanese nationwide database) showed that over half of the cases of pediatricians’ decision-making process errors had an emotional component [23]. A successful management of emotional responses happens only when a healthcare worker (just like any other individual) is able to recognize these responses and integrate them into a clinician’s matrix of professional understanding [24,25,26]. Hence, given that COVID-19 pandemic can provoke the onset or exacerbate symptoms of mental health issues [27,28,29], there is a critical need for research studies that provide a deeper understanding of the mental health status of PHPs during the COVID-19 pandemic, which could identify potential targets for interventions to support these workers and, indirectly, their patients.

Italy was the first European country to be hit hard by the coronavirus and to impose national lockdown measures, which has, among other things, affected the population’s health-related quality of life and disrupted the National Health System (see Table 1, Figure 1). The results of a self-administered survey of 77 Italian perinatal facilities [30] indicated that 70% of them had been negatively influenced in the functioning of one or more aspects of their clinical services by the first wave of the COVID-19 pandemic. Less than one third of facilities continued to provide outpatient routine visits as usual, while the majority of all the facilities continued to be completely (68.8%) or partially (19.7%) available for emergencies. Another key finding was that just under a quarter of facilities became understaffed. Furthermore, the staff of 68.2% of the facilities considered both the use of personal protective equipment and the adoption of social distancing to be very stressful.

In the light of the above, the present study adopted a cross-sectional survey design to assess the anxiety, stress, depression, and burnout status of perinatal health professionals during the COVID-19 pandemic in Italy.

## 2. Materials and Methods

### 2.1. Study Design and Participants

This survey, based on non-probability and a snowball sampling design, was conducted in Italy from June to October 2020. We surveyed perinatal healthcare professionals using an online questionnaire designed by the Observatory of Perinatal Clinical Psychology (https://www.unibs.it/it/node/988 (accessed on 16 May 2021) of the Department of Clinical and Experimental Sciences, Section of Neuroscience (University of Brescia, Italy), administered via the LimeSurvey platform.

The survey questionnaire was distributed by sending an electronic link via the mailing lists of the main Italian perinatal healthcare professional associations and registers. A brief explanation of the study purpose and an assurance of anonymity were outlined in the invitation email as well as on the first page. The research was approved by the Ethics Committee of the ASST Spedali Civili Hospital Brescia on 24 June 2020 (ethical number: NP4221).

### 2.2. Survey Description

The questionnaire consisted of two parts: basic sociodemographic and work data and the mental health assessment. Sociodemographic and work data included, for example, age, gender, professional title, job type and location, and years of work experience. For the mental health assessment, we used two validated scales: the Depression, Anxiety, and Stress Scale-21 (DASS-21) was used to evaluate depressive, anxiety, and stress symptoms; the Maslach Burnout Inventory (MBI) was used to measure burnout symptoms. The DASS-21 [31,32] is a self-reported 21-item scale used to measure levels of depression (7 items), anxiety (7 items), and stress (7 items) symptoms in both clinical and nonclinical samples of adults. Items are scored on a scale of 0 (does not apply to me at all) to 3 (applies to me much or most of the time). The DASS-21 subscales were scored as follows: normal to mild (≤9 points) and moderate to extremely severe (≥10 points) symptomatology. The DASS-21 showed excellent internal consistency (0.93) and sound construct validity. The abbreviated MBI [33,34] is a self-reported measure of burnout. The inventory consists of 9 statements about work-related attitudes and feelings that refer to three dimensions of burnout (personal accomplishment, depersonalization, and emotional exhaustion). The respondents were asked to rate the frequency of particular feelings associated with their job on a 7-point Likert-type scale ranging from 0 (never) to 6 (every day). A high risk of emotional exhaustion (>10 points) along with a high risk of depersonalization (>6 points) or personal accomplishment (<13 points) indicates burnout. The MBI’s internal consistency ranges between 0.70 and 0.80 [17].

### 2.3. Study Size

The survey was based on a snowball sampling design, and it was not driven by any pre-specified hypothesis, being essentially descriptive in nature. For this reason, it was not dimensioned using a formal statistical test. A total of 200 perinatal healthcare workers allowed to estimate a two-sided 95% confidence interval with a precision of 0.055, assuming that 20% of study participants could have developed severe anxiety symptoms, depression symptoms, and perceived stress levels during COVID-19 pandemic.

### 2.4. Statistical Analyses

A complete case analysis approach was used to manage missing data. Categorical variables were reported in terms of frequency, while continuous variables were synthesized in terms of mean, standard deviation, and range. For the purposes of this study, anxiety, stress, and depression, as measured by the DASS-21 scale, were dichotomized by grouping normal and mild levels and moderate to extremely severe levels using the following cutoffs: anxiety (<10 vs. ≥10), stress (≥19 vs. <19), and depression (<14 vs. ≥14). The associations between a priori selected variables and moderate to extremely severe levels of anxiety, stress, and depression were assessed through univariate and multivariate logistic regression models. The results are reported in terms of odds ratios (ORs) and related 95% confidence intervals (CIs). An OR greater than 1 means a higher likelihood of a moderate to extremely severe level of the investigated domain (anxiety, stress, or depression) within the considered explanatory variable. Conversely, an OR less than 1 means a lower likelihood of a moderate to extremely severe level. Analyses were carried out through R version 4.0.2 (R Foundation for Statistical Computing, Vienna, Austria).

## 3. Results

### 3.1. Characteristics of the Sample

This study included 195 PHPs distributed as follows: 59 (30.4%) midwives, 48 (24.7%) psychologists, 30 (15.5%) physicians, and 57 (29.4%) “other” positions (including nurses, nursery nurses, social-health assistants, physiotherapists, psychomotor therapists, and speech therapists). Among the sample, 92.8% were female, and mean age was 44.8 (SD = 11.3). Most PHPs worked in outpatient facilities, including family counseling services (43.0%), outpatient healthcare clinics (15.2%), and private offices (11.5%). The range of work experience varied from less than 1 year to more than 20 years, with 55.4% of PHPs having more than 11 years of work experience. The details of the demographic and work characteristics are presented in Table 2.

### 3.2. Severity and Scores

About one-fifth of the participants scored above the cut-off scores for at least 1 of the 3 DASS-21 subscales. The mean scores for depression, anxiety, and stress were 5.4, 3.6, and 9.6, while 18.7, 18.7, and 21.5% of PHPs had symptoms of depression, anxiety, and perceived stress above the cutoff values, respectively. Regarding related factors in these 3 domains, we found significant differences in symptomatology levels between people who did and did not suffer from trauma not related to the pandemic (*p* < 0.001 for depression and anxiety, *p* = 0.004 for stress). Different levels of stress were also found between different professions (*p* = 0.034) and between those who were infected or not (*p* = 0.030).

The mean MBI score was 23.5. Among the respondents, 6.2% were above the cut-off for burnout, with significant differences among PHPs who continued to work or not during the pandemic (*p* = 0.045) and those who were or were not worried about being infected with COVID-19. It must be noted that regression analyses of burnout were not feasible since only 10 participants experienced burnout.

### 3.3. Independent Risk Factors

The univariate regression analysis (see Table 3) shows that suffering from recent (up to 3 months) trauma unrelated to the pandemic was positively significantly associated with more severe symptoms of depression (OR: 6.62, 95% CI: 2.67–16.47), anxiety (OR: 6.62, 95% CI: 2.67–16.47), and stress (OR: 3.50, 95% CI: 1.40–8.49). It was also found that having more than 20 years of work experience was associated with symptoms of anxiety above the cut-off. Furthermore, being affected by COVID-19 (OR: 4.13, 95% CI: 0.97–16.51) was found to be associated with high perceived stress levels. Lastly, compared to physicians, psychologists and PHPs working in other positions (OR: 0.12, 95% CI: 0.05–0.073; OR: 0.33, 95% CI: 0.10–0.99) showed significantly reduced odds of high perceived stress.

### 3.4. Risk Factors and Psychological Distress

In the multivariable-adjusted regression model (Table 4), suffering from trauma unrelated to the pandemic was associated with higher levels of depression (aOR: 7.34, 95% CI: 2.73–20.28), anxiety (aOR: 6.13, 95% CI: 2.28–16.73), and stress (aOR: 3.20, 95% CI: 1.14–8.88). Compared to working as a physician, working as a psychologist was associated with lower odds of developing depressive symptomatology (aOR: 0.21, 95% CI: 0.04–0.94) and perceived stress (aOR: 0.19, 95% CI: 0.04–0.80). Additionally, comparing working in other positions to working as a physician, the aOR for stress was 0.19 (95% CI: 0.05–0.70).

## 4. Discussion

To the best of our knowledge, this was the first study to examine common psychological morbidities among PHPs during the COVID-19 pandemic. We found that about one-fifth of PHPs had clinically relevant distress symptoms. The estimated self-reported rates of depression (18.7%), anxiety (18.7%), and stress (21.5%) symptoms among our sample (surveyed during the period of June–October 2020) were lower than those reported in the Italian general population [35,36,37] and in Italian general healthcare workers [35,38,39] using the same measurement during the period of March–May 2020. However, our rates are slightly higher than the 14.4% rate of depressive symptoms registered by an Italian population study in June 2020 [40] and notably higher than those reported by pre-pandemic Italian population-based studies, showing prevalence rates of 5.4 and 7% for depressive symptoms and anxiety, respectively [41]. In our study, about 5% of PHPs suffered from burnout, while other Italian studies using the same assessment tool found that just less than one-third of health professionals had burnout during the first three months of the COVID-19 outbreak [35,38]. From these data, we hypothesize that the pandemic increased psychological distress among Italian PHPs, and that after the first three months of the pandemic, their quality of working life was still affected by the global health emergency, but to a lesser extent than at the beginning. This may be due to a certain adjustment and familiarization by the healthcare system and services, as well as adjustments in PHPs’ lifestyles to the new situation. Overall, these data suggest a critical need for psychological assessment of and support for PHPs, as mental health is fundamental for their own health and well-being as well as for their productivity and effectiveness at work, which influence the quality of medical assistance they provide and hence patients’ safety.

This study was also aimed at identifying related factors that endanger PHPs’ psychological health, as these may be fruitful targets for screening and preventive interventions. A recent history of trauma not related to COVID-19 was one such factor. Indeed, suffering from trauma not related to the pandemic was associated with more severe symptoms of depression, anxiety, and stress. This result aligns with the findings from a previous COVID-19 study, indicating that having a history of stressful situations was associated with higher levels of depression and anxiety [36]. Likely, pandemic-related adverse psychological effects are more severe and long-lasting because of a cumulative effect of traumatic experiences [42]. This hypothesis is in line with the literature on psychological sequelae of trauma, which highlights that individuals recently exposed to a trauma are at greater risk of experiencing psychological symptoms when facing situations of uncertainty [43].

It is also notable that the professional role was related to depressive symptoms and perceived stress; more specifically, physicians were at a higher risk compared with midwives and, especially, psychologists and other healthcare professionals. This is consistent with another COVID-19 study, which found that physicians were at higher risk for severe to extremely severe depressive symptoms compared with non-physicians [44]. A recent study on the impact of the pandemic on perinatal healthcare services in Italy over the period of March–May 2020 gives us a possible explanation for the differences among professions, reporting that most facilities continued to provide in-person visits with physicians (82.6%), but not with psychologists (32.8%), who have made extensive use of telepsychology. It is thus reasonable to hypothesize that telehealth may not only be an important tool in prevention, diagnosis, treatment, and control regarding patients’ physical and psychological health, while keeping patients and healthcare workers safe and minimizing the risk of COVID-19 transmission [45,46], but it may also be helpful in protecting healthcare workers’ mental health.

With regards to the possible practical implications of this study for health policy-makers and health system managers, our findings suggest the clinical importance that PHPs have (a) easy and regular access to screening services (including online screening tools, e.g., www.hgaps.org/assessment-center.html (accessed on 16 May 2021), and (b) free access to psychological help services specifically dedicated to healthcare professionals.

The strengths of this study include the focus on a specific and seldom studied population of healthcare workers (PHPs), the enrollment of PHPs representing different professional roles, and the use of validated questionnaires to measure psychological outcomes. However, there were several limitations. First, this was a cross-sectional study, which did not allow us to distinguish between old and new symptoms or to identify the symptom trajectories throughout the pandemic phases. Second, the study used a non-probability and snowball sampling survey instead of random sampling; thus, our sample may not be representative of the overall population. Third, because of the restrictive measures taken to contain the pandemic, we opted for an online survey to access PHPs, a data collection method that may have further limited the generalizability of the results due to the non-response bias (which could not be assessed in the present study). Additionally, the difference between online and face-to-face surveys should also be kept in mind. Fourth, we had to merge PHPs other than physicians, midwives, and psychologists into the category “other healthcare professionals” to obtain statistical power, even though some professionals, such as nurses, are likely to be more vulnerable to the pandemic emergency than others, such as speech therapists. Finally, we do not have information about the specific type and exact time of the non-pandemic traumas.

## 5. Conclusions

In this study of PHPs working during the COVID-19 pandemic in Italy, high rates of self-reported symptoms of depression and anxiety and perceived stress were reported between the first and second wave of the pandemic. Suffering from trauma unrelated to the pandemic was associated with higher levels of psychological distress. Considering our findings and the detrimental effects of psychological distress on healthcare workers’ quality of personal and work life, as well as patient care and professional efficiency [47], providing timely and regular evidence-based assessments [46,48] of psychological distress targeting PHPs (with particular attention reserved for physicians) appears to be of primary importance to reduce their burden at the individual health level and to improve their provision of health care. Hence, health authorities should immediately implement and integrate such assessment into their work plans.

## Figures and Tables

**Figure 1 ijerph-18-06542-f001:**
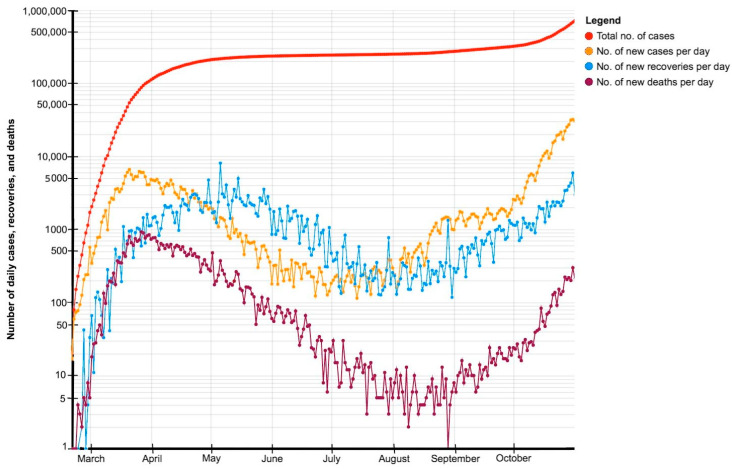
Development of the pandemic in Italy from 21 February to 31 October 2020. Note: Figure adapted from the article “Statistics of the COVID-19 pandemic in Italy” (available online at https://en.wikipedia.org/wiki/Statistics_of_the_COVID-19_pandemic_in_Italy (accessed on 16 May 2021).

**Table 1 ijerph-18-06542-t001:** Timeline of the COVID-19 pandemic in Italy.

Time Period	Main Events Related to COVID-19
31 December 2019	Wuhan Municipal Health Commission in Wuhan City, Hubei Province, China, reports a cluster of pneumonia cases (including seven severe cases) of unknown etiology.
9 January 2020	China CDC reports that a novel coronavirus (later named SARS-CoV-2, the virus causing COVID-19) was detected as the causative agent of 15 of 59 cases of pneumonia.
17 January 2020	ECDC publishes its first risk assessment on the novel coronavirus.
22 January 2020	Italian Ministry of Health instructs a task force to coordinate a surveillance system for suspected cases and interventions in national territory.
30 January 2020	Two Chinese tourists hospitalized for respiratory tract infections are the first confirmed cases of COVID-19 detected in Italy. WHO declares this first outbreak a “public health emergency of international concern.”
31 January 2020	Italian Council of Ministers declares a national public health emergency.
21 February 2020	Italian National Institute of Health confirms the first case of local transmission of COVID-19 infection. Over the following days, Italian authorities report clusters of cases in several regions (Lombardy, Piedmont, Veneto, etc.).
8–9 March 2020	Italian Council of Ministers issues a decree to install strict public health measures starting in the most affected regions (Lombardy and Veneto), including social distancing and restricting movements of people within and outside hometowns, with permitted travel limited to shopping for food, going to work (only for essential services to remain operating; work from home was encouraged), or seeking medical care. All planned surgeries are postponed in order to give intensive care beds over to the treatment of COVID-19 patients.
11 March 2020	WHO Director General declares COVID-19 a “global pandemic.”Italian Council of Ministers extends strict containment measures at national level.
13 March 2020	WHO declares that Europe is becoming the new epicenter of the COVID-19 pandemic.
31 March 2020	Italian Ministry of Health issues recommendations for pregnant women in labor, puerperal women, newborns, and breastfeeding mothers.
April 2020	Italian scientific associations in the field of perinatal medicine (e.g., FIGO and SIN) start publishing interim recommendations for management of pregnant women in labor, puerperal women, newborns, and breastfeeding mothers during the COVID-19 pandemic.
4 May 2020	Italian Council of Ministers restores freedom of movement, and other not essential activities are re-opened later in the month.
18 May 2020	Most businesses reopened and free movement within region of residence granted to all citizens, while movement across regions is still banned for non-essential purposes.
31 May 2020	Istituto Superiore di Sanità (in collaboration with ACP, AGUI, AOGOI, FNOPO, SIAARTI, SIGO, SIMP, SIN, SIP, and TAS) publishes interim indications for pregnancy, childbirth, breastfeeding, and the care of children 0–2 years in response to the COVID-19 emergency.
3 June 2020	Free movement within Italian national territory restored.
October 2020	Italy hit by second wave of the pandemic. Italian Council of Ministers postpones the end of the state of emergency to 31 January 2021, and it reintroduces stricter rules to limit the spread of COVID-19.

Note: Table adapted from the timeline of ECDC’s response to COVID-19 (available online at www.ecdc.europa.eu/en/covid-19/timeline-ecdc-response (accessed on 16 May 2021). ACP, Associazione Culturale Pediatri; AOGOI, Associazione Ostetrici Ginecologi Ospedalieri Italiani; AGUI, Associazione Ginecologi Universitari Italiani; China CDC, Chinese Center for Disease Control and Prevention; ECDC, European Centre for Disease Prevention and Control; FIGO, International Federation of Gynecology and Obstetrics; FNOPO, Federazione Nazionale degli Ordini della Professione di Ostetrica; SIAARTI, Società Italiana di Anestesia Analgesia Rianimazione e Terapia Intensiva; SIGO, Società Italiana di Ginecologia e Ostetricia; SIMP, Società Italiana di Medicina Perinatale; SIN, Società Italiana di Neonatologia; SIP, Società Italiana di Pediatria; TAS, Tavolo Tecnico Allattamento del Ministero della Salute; WHO, World Health Organization.

**Table 2 ijerph-18-06542-t002:** Demographic and occupational characteristics of perinatal healthcare professionals, and prevalence of depression, anxiety, stress, and burnout.

Characteristic	Study Sample	Stress			Anxiety			Depression			Burnout ^a^		
	*N* = 194 (100%)	No	Yes	*p*	No	Yes	*p*	No	Yes	*p*	No	Yes	*p*
Working sector				0.971			0.200			0.635			0.233
Antenatal	13 (6.7%)	11 (6.9%)	2 (5.9%)		9 (5.5%)	4 (13.3%)		10 (6.1%)	3 (10.0%)		10 (6.1%)	0 (0.0%)	
Postnatal	55 (28.4%)	45 (28.1%)	10 (29.4%)		49 (29.9%)	6 (20.0%)		48 (29.3%)	7 (23.3%)		49 (30.1%)	1 (10.0%)	
Both	126 (64.9%)	104 (65.0%)	22 (64.7%)		106 (64.6%)	20 (66.7%)		106 (64.6%)	20 (66.7%)		104 (63.8%)	9 (90.0%)	
Age				0.086			0.089			0.271			0.052
Mean (SD)	44.722 (11.271)	45.362 (10.863)	41.706 (12.770)		45.311 (11.330)	41.500 (10.550)		45.104 (11.151)	42.633 (11.886)		44.957 (11.191)	37.800 (11.793)	
Range	24.000–66.000	26.000–66.000	24.000–65.000		24.000–66.000	25.000–60.000		24.000–66.000	25.000–65.000		24.000–66.000	25.000–65.000	
Gender				0.289			0.371			0.899			0.123
Male	14 (7.2%)	13 (8.1%)	1 (2.9%)		13 (7.9%)	1 (3.3%)		12 (7.3%)	2 (6.7%)		11 (6.7%)	2 (20.0%)	
Female	180 (92.8%)	147 (91.9%)	33 (97.1%)		151 (92.1%)	29 (96.7%)		152 (92.7%)	28 (93.3%)		152 (93.3%)	8 (80.0%)	
Region				0.926			0.591			0.309			0.779
South	9 (4.6%)	7 (4.4%)	2 (5.9%)		7 (4.3%)	2 (6.7%)		6 (3.7%)	3 (10.0%)		6 (3.7%)	0 (0.0%)	
North	155 (79.9%)	128 (80.0%)	27 (79.4%)		130 (79.3%)	25 (83.3%)		132 (80.5%)	23 (76.7%)		132 (81.0%)	8 (80.0%)	
Center	30 (15.5%)	25 (15.6%)	5 (14.7%)		27 (16.5%)	3 (10.0%)		26 (15.9%)	4 (13.3%)		25 (15.3%)	2 (20.0%)	
Working position				0.193			0.569			0.619			0.531
Temporarily employed	7 (3.6%)	4 (2.5%)	3 (8.8%)		5 (3.0%)	2 (6.7%)		5 (3.0%)	2 (6.7%)		7 (4.3%)	0 (0.0%)	
Employed	148 (76.3%)	124 (77.5%)	24 (70.6%)		125 (76.2%)	23 (76.7%)		126 (76.8%)	22 (73.3%)		122 (74.8%)	9 (90.0%)	
Freelancer	39 (20.1%)	32 (20.0%)	7 (20.6%)		34 (20.7%)	5 (16.7%)		33 (20.1%)	6 (20.0%)		34 (20.9%)	1 (10.0%)	
Professional role				0.034			0.236			0.280			0.341
Physician	30 (15.5%)	21 (13.1%)	9 (26.5%)		25 (15.2%)	5 (16.7%)		23 (14.0%)	7 (23.3%)		22 (13.5%)	3 (30.0%)	
“Other” position	57 (29.4%)	50 (31.2%)	7 (20.6%)		46 (28.0%)	11 (36.7%)		49 (29.9%)	8 (26.7%)		52 (31.9%)	2 (20.0%)	
Midwifery	59 (30.4%)	45 (28.1%)	14 (41.2%)		48 (29.3%)	11 (36.7%)		48 (29.3%)	11 (36.7%)		49 (30.1%)	4 (40.0%)	
Psychologist	48 (24.7%)	44 (27.5%)	4 (11.8%)		45 (27.4%)	3 (10.0%)		44 (26.8%)	4 (13.3%)		40 (24.5%)	1 (10.0%)	
Workplace				0.212			0.197			0.071			0.364
Missing data	29	23	6		24	5		26	3		23	4	
Outpatient clinic	25 (15.2%)	20 (14.6%)	5 (17.9%)		18 (12.9%)	7 (28.0%)		20 (14.5%)	5 (18.5%)		19 (13.6%)	2 (33.3%)	
Family counseling	71 (43.0%)	62 (45.3%)	9 (32.1%)		63 (45.0%)	8 (32.0%)		64 (46.4%)	7 (25.9%)		65 (46.4%)	2 (33.3%)	
Ward of obstetrics and gynecology	31 (18.8%)	23 (16.8%)	8 (28.6%)		26 (18.6%)	5 (20.0%)		21 (15.2%)	10 (37.0%)		22 (15.7%)	2 (33.3%)	
Neonatal intensive care unit	19 (11.5%)	18 (13.1%)	1 (3.6%)		18 (12.9%)	1 (4.0%)		17 (12.3%)	2 (7.4%)		19 (13.6%)	0 (0.0%)	
Private practice	19 (11.5%)	14 (10.2%)	5 (17.9%)		15 (10.7%)	4 (16.0%)		16 (11.6%)	3 (11.1%)		15 (10.7%)	0 (0.0%)	
Work experience (years)				0.284			0.144			0.987			0.421
<1	8 (4.1%)	5 (3.1%)	3 (8.8%)		5 (3.0%)	3 (10.0%)		6 (3.7%)	2 (6.7%)		7 (4.3%)	0 (0.0%)	
1–5	46 (23.7%)	39 (24.4%)	7 (20.6%)		41 (25.0%)	5 (16.7%)		39 (23.8%)	7 (23.3%)		38 (23.3%)	5 (50.0%)	
6–10	33 (17.0%)	24 (15.0%)	9 (26.5%)		27 (16.5%)	6 (20.0%)		28 (17.1%)	5 (16.7%)		26 (16.0%)	1 (10.0%)	
11–15	29 (14.9%)	26 (16.2%)	3 (8.8%)		22 (13.4%)	7 (23.3%)		25 (15.2%)	4 (13.3%)		25 (15.3%)	2 (20.0%)	
16–20	19 (9.8%)	17 (10.6%)	2 (5.9%)		15 (9.1%)	4 (13.3%)		16 (9.8%)	3 (10.0%)		15 (9.2%)	1 (10.0%)	
>20	59 (30.4%)	49 (30.6%)	10 (29.4%)		54 (32.9%)	5 (16.7%)		50 (30.5%)	9 (30.0%)		52 (31.9%)	1 (10.0%)	
Working during pandemic				0.109			0.239			0.960			0.045
Missing data	1	1	0		0	1		1	0		1	0	
As usual	79 (40.9%)	67 (42.1%)	12 (35.3%)		67 (40.9%)	12 (41.4%)		68 (41.7%)	11 (36.7%)		69 (42.6%)	2 (20.0%)	
More than usual	58 (30.1%)	43 (27.0%)	15 (44.1%)		47 (28.7%)	11 (37.9%)		48 (29.4%)	10 (33.3%)		45 (27.8%)	7 (70.0%)	
Less than usual	50 (25.9%)	45 (28.3%)	5 (14.7%)		46 (28.0%)	4 (13.8%)		42 (25.8%)	8 (26.7%)		44 (27.2%)	1 (10.0%)	
Did not work because of the pandemic	6 (3.1%)	4 (2.5%)	2 (5.9%)		4 (2.4%)	2 (6.9%)		5 (3.1%)	1 (3.3%)		4 (2.5%)	0 (0.0%)	
Transferred to another ward or department				0.764			0.819			0.819			0.314
Yes	19 (9.8%)	15 (9.4%)	4 (11.8%)		17 (10.4%)	2 (6.7%)		17 (10.4%)	2 (6.7%)		16 (9.8%)	0 (0.0%)	
No	157 (80.9%)	131 (81.9%)	26 (76.5%)		132 (80.5%)	25 (83.3%)		132 (80.5%)	25 (83.3%)		132 (81.0%)	10 (100%)	
Working in private office	18 (9.3%)	14 (8.8%)	4 (11.8%)		15 (9.1%)	3 (10.0%)		15 (9.1%)	3 (10.0%)		15 (9.2%)	0 (0.0%)	
Working in a COVID-19 unit				0.848			0.805			0.805			0.582
Yes	11 (5.7%)	9 (5.6%)	2 (5.9%)		10 (6.1%)	1 (3.3%)		10 (6.1%)	1 (3.3%)		10 (6.1%)	0 (0.0%)	
No	8 (4.1%)	6 (3.8%)	2 (5.9%)		7 (4.3%)	1 (3.3%)		7 (4.3%)	1 (3.3%)		6 (3.7%)	0 (0.0%)	
Answered negatively to the previous item	175 (90.2%)	145 (90.6%)	30 (88.2%)		147 (89.6%)	28 (93.3%)		147 (89.6%)	28 (93.3%)		147 (90.2%)	10 (100%)	
New working role/task because of pandemic				0.565			0.598			0.755			0.517
Yes	30 (15.5%)	24 (15.0%)	6 (17.6%)		27 (16.5%)	3 (10.0%)		24 (14.6%)	6 (20.0%)		21 (12.9%)	1 (10.0%)	
No	144 (74.2%)	121 (75.6%)	23 (67.6%)		121 (73.8%)	23 (76.7%)		123 (75.0%)	21 (70.0%)		125 (76.7%)	9 (90.0%)	
Not applicable	20 (10.3%)	15 (9.4%)	5 (14.7%)		16 (9.8%)	4 (13.3%)		17 (10.4%)	3 (10.0%)		17 (10.4%)	0 (0.0%)	
Recent history of pandemic-unrelated trauma				0.004			<0.001			<0.001			0.433
Yes	27 (13.9%)	17 (10.6%)	10 (29.4%)		15 (9.1%)	12 (40.0%)		15 (9.1%)	12 (40.0%)		19 (11.7%)	2 (20.0%)	
No	167 (86.1%)	143 (89.4%)	24 (70.6%)		149 (90.9%)	18 (60.0%)		149 (90.9%)	18 (60.0%)		144 (88.3%)	8 (80.0%)	
Having been infected by SARS-CoV-2				0.030			0.566			0.566			0.325
Yes	9 (4.6%)	5 (3.1%)	4 (11.8%)		7 (4.3%)	2 (6.7%)		7 (4.3%)	2 (6.7%)		6 (3.7%)	1 (10.0%)	
No	185 (95.4%)	155 (96.9%)	30 (88.2%)		157 (95.7%)	28 (93.3%)		157 (95.7%)	28 (93.3%)		157 (96.3%)	9 (90.0%)	
Fear of becoming infected with SARS-CoV-2				0.188			0.536			0.981			0.024
Little or no	87 (44.8%)	76 (47.5%)	11 (32.4%)		76 (46.3%)	11 (36.7%)		74 (45.1%)	13 (43.3%)		76 (46.6%)	2 (20.0%)	
Neither yes nor no	45 (23.2%)	37 (23.1%)	8 (23.5%)		38 (23.2%)	7 (23.3%)		38 (23.2%)	7 (23.3%)		40 (24.5%)	1 (10.0%)	
Quite-to-very worried	62 (32.0%)	47 (29.4%)	15 (44.1%)		50 (30.5%)	12 (40.0%)		52 (31.7%)	10 (33.3%)		47 (28.8%)	7 (70.0%)	

^a^ Out of the 194 participants, only 173 completed the Maslach Burnout Inventory.

**Table 3 ijerph-18-06542-t003:** Associations between demographic and occupational variables and stress, anxiety, and depression.

Predictors	Stress		Anxiety		Depression	
	OR (95% CI)	*p*	OR (95% CI)	*p*	OR (95% CI)	*p*
Working sector (ref. Antenatal)						
Postnatal	1.22 (0.27–8.69)	0.812	0.42 (0.12–1.69)	0.186	0.49 (0.11–2.55)	0.351
Both	1.16 (0.29–7.86)	0.851	0.28 (0.06–1.25)	0.082	0.63 (0.17–2.99)	0.509
Age (per 1 year increase)	0.97 (0.94–1.00)	0.088	0.97 (0.93–1.00)	0.091	0.98 (0.95–1.02)	0.270
Gender (ref. Male)						
Female	2.92 (0.55–53.93)	0.310	2.50 (0.47–46.22)	0.387	1.11 (0.28–7.35)	0.899
Region (ref. South)						
North	0.74 (0.17–5.14)	0.714	0.67 (0.15–4.69)	0.634	0.35 (0.09–1.74)	0.156
Center	0.70 (0.12–5.61)	0.704	0.39 (0.05–3.38)	0.348	0.31 (0.05–1.89)	0.184
Working position (ref. Temporarily employed)						
Employed	0.26 (0.05–1.38)	0.089	0.46 (0.09–3.35)	0.370	0.44 (0.09–3.18)	0.340
Freelancer	0.29 (0.05–1.75)	0.157	0.37 (0.06–3.04)	0.299	0.45 (0.08–3.68)	0.405
Professional role (ref. Physician)						
“Other” position	0.33 (0.10–0.99)	**0.049**	1.20 (0.39–4.15)	0.763	0.54 (0.17–1.70)	0.280
Midwifery	0.73 (0.27–1.99)	0.524	1.15 (0.37–3.97)	0.818	0.75 (0.26–2.28)	0.603
Psychologist	0.21 (0.05–0.73)	**0.018**	0.33 (0.06–1.47)	0.155	0.30 (0.07–1.09)	0.075
Workplace (ref. Outpatient clinic)						
Family counseling	0.58 (0.18–2.07)	0.376	0.33 (0.10–1.04)	0.055	0.44 (0.13–1.62)	0.196
Ward of obstetrics and gynecology	1.39 (0.40–5.25)	0.610	0.49 (0.13–1.79)	0.287	1.90 (0.57–7.03)	0.307
Neonatal intensive care unit	0.22 (0.01–1.55)	0.188	0.14 (0.01–0.92)	0.082	0.47 (0.06–2.50)	0.402
Private office	1.43 (0.34–6.07)	0.621	0.69 (0.15–2.73)	0.599	0.75 (0.14–3.54)	0.720
Work experience (years) (ref. < 1)						
1–5	0.30 (0.06–1.71)	0.150	0.20 (0.04–1.22)	0.067	0.54 (0.10–4.17)	0.498
6–10	0.63 (0.12–3.54)	0.570	0.37 (0.07–2.19)	0.247	0.54 (0.09–4.35)	0.511
11–15	0.19 (0.03–1.29)	0.083	0.53 (0.10–3.10)	0.455	0.48 (0.07–4.04)	0.453
16–20	0.20 (0.02–1.49)	0.119	0.44 (0.07–2.89)	0.379	0.56 (0.07–5.07)	0.577
>20	0.34 (0.07–1.87)	0.182	0.15 (0.03–0.92)	**0.031**	0.54 (0.10–4.08)	0.490
Working during pandemic (ref. As usual)						
More than usual	1.95 (0.83–4.63)	0.124	1.31 (0.53–3.23)	0.560	1.29 (0.50–3.29)	0.595
Less than usual	0.62 (0.19–1.80)	0.399	0.49 (0.13–1.49)	0.235	1.18 (0.42–3.15)	0.746
Did not work because of the pandemic	2.79 (0.36–16.07)	0.265	2.79 (0.36–16.07)	0.265	1.24 (0.06–8.69)	0.853
Transferred to another ward or department (ref. No)						
Yes	1.34 (0.36–4.06)	0.624	0.62 (0.09–2.35)	0.541	0.62 (0.09–2.35)	0.541
Working in private office	1.44 (0.38–4.39)	0.548	1.06 (0.23–3.50)	0.935	1.06 (0.23–3.50)	0.935
Working in a COVID-19 unit (ref. No)						
Yes	0.67 (0.06–6.84)	0.720	0.70 (0.02–19.75)	0.812	0.70 0.02–19.75)	0.812
Answered negatively to the previous item	0.62 (0.14–4.38)	0.571	1.33 (0.22–25.43)	0.792	1.33 0.22–25.43)	0.792
New working role or task because of pandemic (ref. No)						
Yes	1.32 (0.45–3.41)	0.591	0.58 (0.13–1.84)	0.409	1.46 (0.50–3.84)	0.458
Not applicable	1.75 (0.53–5.04)	0.320	1.32 (0.35–3.98)	0.650	1.03 (0.23–3.42)	0.961
Recent history of pandemic-unrelated trauma (ref. No)						
Yes	3.50 (1.40–8.49)	**0.006**	6.62 (2.67–16.47)	**<0.001**	6.62 (2.67–16.47)	**<0.001**
Having been infected by SARS-CoV-2 (ref. No)						
Yes	4.13 (0.97–16.51)	**0.043**	1.60 (0.23–7.04)	0.569	1.60 (0.23–7.04)	0.569
Fear of becoming infected with SARS-CoV-2 (ref. Little or no)						
Neither yes nor no	1.49 (0.54–4.01)	0.428	1.27 (0.44–3.50)	0.645	1.05 (0.37–2.79)	0.926
Quite-to-very worried	2.21 (0.94–5.32)	0.071	1.66 (0.68–4.10)	0.267	1.09 (0.44–2.68)	0.843

Note: OR = odds ratio. Bold *p*-values indicate statistical significance.

**Table 4 ijerph-18-06542-t004:** Adjusted odds ratios and confidence intervals of the associations with stress, anxiety, and depression.

Predictors	Stress		Anxiety		Depression	
	aOR (95% CI)	*p*	aOR (95% CI)	*p*	aOR (95% CI)	*p*
Age (per 1 year increase)	0.97 (0.93–1.01)	0.152	0.98 (0.94–1.02)	0.224	0.99 (0.95–1.03)	0.498
Gender (ref. Male)						
Female	2.80 (0.45–55.28)	0.356	2.06 (0.33–40.74)	0.518	1.00 (0.21–7.54)	0.996
Professional role (ref. Physician)						
“Other” position	0.19 (0.05–0.70)	**0.015**	0.78 (0.20–3.16)	0.712	0.34 (0.08–1.27)	0.110
Midwifery	0.44 (0.14–1.42)	0.167	0.71 (0.19–2.87)	0.610	0.60 (0.17–2.13)	0.418
Psychologist	0.19 (0.04–0.80)	**0.028**	0.28 (0.04–1.52)	0.147	0.21 (0.04–0.94)	**0.046**
Working during pandemic (ref. As usual)						
More than usual	1.58 (0.61–4.12)	0.341	1.05 (0.38–2.85)	0.928	1.01 (0.35–2.84)	0.990
Less than usual	0.80 (0.22–2.60)	0.722	0.49 (0.11–1.70)	0.285	1.52 (0.49–4.59)	0.461
Did not work because of the pandemic	3.48 (0.37–24.44)	0.224	1.93 (0.19–14.48)	0.541	1.00 (0.04–8.85)	0.997
Recent history of pandemic-unrelated trauma (ref. No)						
Yes	3.20 (1.14–8.88)	**0.025**	6.13 (2.28–16.73)	**<0.001**	7.34 (2.73–20.28)	**<0.001**
Having been infected by SARS-CoV-2 (ref. No)						
Yes	3.08 (0.59–16.53)	0.174	0.56 (0.03–3.85)	0.614	1.61 (0.20–8.87)	0.605
Fear of becoming infected with SARS-CoV-2 (ref. No)						
Neither yes nor no	1.43 (0.46–4.27)	0.529	1.11 (0.32–3.54)	0.866	1.14 (0.35–3.39)	0.823
Quite-to-very worried	1.96 (0.76–5.19)	0.168	1.35 (0.49–3.72)	0.557	0.82 (0.30–2.20)	0.700
Observations	193		193		193	
R^2^ Tjur	0.167		0.149		0.149	

Note: aOR = adjusted odds ratio. Bold *p*-values indicate statistical significance.

## Data Availability

The complete dataset is available from the corresponding author upon reasonable request.

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
