# Peer review of "Mental Health States Experienced by Perinatal Healthcare Workers during COVID-19 Pandemic in Italy"

_ijerph, 2021, doi:10.3390/ijerph18126542_

Round 1
Reviewer 1 Report
Thank you for the opportunity to review this paper. The study involved an online questionnaire consisting of items from the DASS-21 to measure symptoms of depression, anxiety, and stress; and the MBI to measure burnout. The sample included 195 perinatal healthcare professionals during the COVID-19 pandemic in Italy. Results of the DASS-21 showed 18.7% reported symptoms of depression, 18.7% reported symptoms of anxiety, and 21.5% reported significant stress. Results of the MBI showed that 6.2% reported burnout. This brief research report provides simple descriptive information that is helpful for understanding the prevalence of mental health issues among PHP’s during this significant time of stress in the country.
I only have 2 questions:
What do “other” professional roles include?
How was missing data handled?
Overall, this is a well-written brief research report.
Author Response
Reviewer 1
1) What do “other” professional roles include?
As we explained in the first paragraph of the Results section, “other” professional roles include: nurses, nursery nurses, social-health assistants, physiotherapists, psychomotor therapists, and speech therapists.
2) How was missing data handled?
A complete case analysis approach was used to manage missing data. The latter are reported in the Tables (see for example Workplace variable in Table 2).
Reviewer 2 Report
The studies of health care professionals and their reactions to COVD-19 pandemic are conducted extensively. The Authors claim that their study is the first one with perinatal health care professionals. This very fact is not enough to conduct the study. There is no other explanation in the text to justify selection of such group. Thus it is not clear what is so unique in the situation of those working in perinatal health care that separate study is required. Moreover the group is mixed and persons with very different experiences and professional role are combined. This strategy is not well explained and justified. The most interesting finding concerns the role of non-pandemic trauma. It would be interesting to know more on types of trauma, the time that has passed since traumatic experience and how this might affect response to COVID-19 pandemic.
In the text the information on calculation of statistical power is not provided although lack of such power (line 145) is used as explanation for lack of certain data analyses or certain data manipluation (line 230 onwards). The abstract (line 21) refers to ORa while is should be OR only (as in the rest of the text).
I would suggest the Authors to clarify the rationale for selection of such study sample and to include some comments on how situation of healthcare staff working in perinatal setting might be linked to their reactions to pandemic.
Author Response
Reviewer 2
1) In the text the information on calculation of statistical power is not provided although lack of such power (line 145) is used as explanation for lack of certain data analyses or certain data manipulation (line 230 onwards). The abstract (line 21) refers to ORa while is should be OR only (as in the rest of the text).
A Study size section (paragraph 2.3) has been added. With regards to the abstract, the ones reported are adjusted odds ratio; see Result's paragraph" 3.4. Risk factors and psychological distress" and Table 4 (now better renamed "Adjusted odds ratios and confidence intervals of the associations with stress, anxiety, and depression").
2) I would suggest the Authors to clarify the rationale for selection of such study sample and to include some comments on how situation of healthcare staff working in perinatal setting might be linked to their reactions to pandemic.
Thank you for this suggestion, it allowed to improve the "power" of the entire paper. We added a specific paragraph in the Introduction section.
Reviewer 3 Report
This is an interesting and important paper looking at psychological distress symptoms and burnout among perinatal healthcare professionals (PHPs) during the Covid-19 pandemic in Italy. Although I think this paper should be published, I would suggest authors some revisions, as follows:
INTRODUCTION
The introduction is well written. However, it could be useful for readers to understand better why it is important to explore mental health of PHPs during the Covid-19 pandemic. Thus, I suggest authors to address the following points:
- Potential differences in terms of overload, mental health, and burnout between PHPs and other healthcare professionals in general and during the Covid-19 outbreak.
- What happened in Italy regarding perinatal healthcare during the Covid-19 pandemic? What has changed since the period before the pandemic?
- Is there a difference between the first period of the pandemic (i.e., the first lockdown) and the period in which the data were collected (i.e., the second wave of the pandemic)?
METHODS
- It is not clear how participants were recruited? Please, describe the channels used to recruit them.
- Please, describe how the study was presented to participants.
- Why such a low rate of men?
- Authors should report information about handling of missing data and potential outliers.
RESULTS
- The results are well-written.
DISCUSSION
- I would suggest authors to report some implications of their study in the field of public health and clinical psychology. For instance, why is it important to help this specific professional category in psychological terms? What could happen to patients accessing this healthcare area if professionals have health and burnout problems?
Author Response
Reviewer 3
Introduction
1) Potential differences in terms of overload, mental health, and burnout between PHPs and other healthcare professionals in general and during the Covid-19 outbreak.
2) What happened in Italy regarding perinatal healthcare during the Covid-19 pandemic? What has changed since the period before the pandemic?
3) Is there a difference between the first period of the pandemic (i.e., the first lockdown) and the period in which the data were collected (i.e., the second wave of the pandemic)?
Thank you for these suggestions that, along with the one by the Reviewer 2, enabled us to make the introduction stronger. We added two new paragraphs in the Introduction section.
Methods
1) It is not clear how participants were recruited? Please, describe the channels used to recruit them.
We have specified that the survey was distributed by sending an electronic link via the mailing lists of Italian perinatal healthcare professional associations and registers.
2) Please, describe how the study was presented to participants.
A brief explanation of the study purpose and an assurance of anonymity were outlined in the invitation email as well as on the first page.
3) Why such a low rate of men?
We don't know the reason. Likely such a low rate is due to the smaller percentage of men (vs women) who work as midwives, psychologists, nurses, and nursery nurses, social-health assistants in Italy.
4) Authors should report information about handling of missing data and potential outliers.
A complete case analysis approach was used to manage missing data.
Discussion
I would suggest authors to report some implications of their study in the field of public health and clinical psychology. For instance, why is it important to help this specific professional category in psychological terms? What could happen to patients accessing this healthcare area if professionals have health and burnout problems?
The implication for the patients are discussed in the Introduction section. With regards to the possible practical implications of our study for health policy‐makers and health system managers, we wrote that our findings suggest the clinical importance that perinatal workers would have easy and regular access to screening services, and free access psychological help service specifically dedicated to healthcare professionals.
Round 2
Reviewer 2 Report
The Authors have sufficiently addressed the issues included in my previous review and, additionally, extended the list of references. The provided explantions and amendements make the manusript better in comparison to the first version.